# Heat Treatment Enhances the Neuroprotective Effects of Crude Ginseng Saponin by Increasing Minor Ginsenosides

**DOI:** 10.3390/ijms24087223

**Published:** 2023-04-13

**Authors:** Yun-Jeong Ji, Hyung Don Kim, Eun Suk Lee, Gwi Yeong Jang, Hyun-A Seong

**Affiliations:** 1Department of Herbal Crop Research, National Institute of Horticultural Herbal Science, Rural Development Administration, Eumseong 27709, Republic of Korea; jyj2842@korea.kr (Y.-J.J.);; 2Department of Biochemistry, School of Life Sciences, Chungbuk National University, Cheongju 28644, Republic of Korea

**Keywords:** *Panax ginseng*, crude saponin, ginsenoside composition, glutamate, heat treatment, neurodegenerative disorders, PC12 cells

## Abstract

Ginsenoside is the primary active substance of ginseng and has many pharmacological effects, such as anti-cancer, immune, regulating sugar and lipid metabolism, and antioxidant effects. It also protects the nervous and cardiovascular systems. This study analyzes the effects of thermal processing on the bioactivities of crude ginseng saponin. Heat treatment increased the contents of minor ginsenosides in crude saponins, such as Rg3, and heat-treated crude ginseng saponin (HGS) had better neuroprotective effects than non-treated crude saponin (NGS). HGS reduced glutamate-induced apoptosis and reactive oxygen species generation in pheochromocytoma 12 (PC12) cells, significantly more than NGS. HGS protected PC12 cells against glutamate-induced oxidative stress by upregulating Nrf2-mediated antioxidant signaling and downregulating MAPK-mediated apoptotic signaling. HGS has the potential for the prevention and treatment of neurodegenerative disorders, such as Alzheimer’s and Parkinson’s disease.

## 1. Introduction

Ginseng (*Panax ginseng* C.A. Meyer.) is a traditional herbal medicine that has been used in Asia for a long time. The pharmacological effects of the physiologically active ingredients of ginseng have been scientifically recognized, and the demand for this root is gradually increasing [1,2]. Ginseng has many therapeutic effects, including on the central nervous system, cardiovascular system, endocrine system, and immune system; it also exhibits anti-tumor, anti-stress, and antioxidant activities [3,4,5,6]. The most important bioactive components of ginseng are ginsenosides, polyacetylenes, polysaccharides, alkaloids, and phenolic compounds. Ginsenosides are the main active ingredients responsible for the therapeutic and pharmacological effects of ginseng, and may be beneficial in cardiovascular diseases and some degenerative diseases. The efficacy and mechanism of action of the ginsenosides of ginseng in Alzheimer’s disease are being actively studied; anti-inflammatory studies of Parkinson’s disease patients are also ongoing [7,8]. The major ginsenosides and active ingredients of ginseng are Rb2, Rb1, Re, Rg1, and Rc; they all aid in the prevention of neurodegenerative diseases [9]. In animal experiments, the ginsenoside Rg1 significantly reduced brain amyloid-beta, and the ginsenosides Rb1, Rc, and Rg5 attenuated glutamic-acid-induced neuronal apoptosis in an in vitro Huntington’s model through the inhibition of Ca^2+^ signaling [10,11]. The ginsenoside Rb1 reduced cerebral edema and improved neurobehavioral function [12]. It is becoming increasingly clear that ginsenosides exert neuroprotective effects by reducing free radical production and enhancing brain function, which highlights their potential to treat neurodegenerative diseases.

Depending on the structure, ginsenosides are divided into protopanaxadiols (PPD), including Rb1, Rb2, Rb3, Rc, Rd, Rg3, and Rh2; protopanaxatriols (PPT), including Re, Rf, Rg1, and Rg2; and oleananes. Ginsenosides differ in physiological activities and bioavailability. Different ways to identify and characterize ginsenosides are being studied in an effort to improve their bioavailability [13].

In traditional oriental medicine, heat treatment has been used to reduce the toxicity of medicinal materials or minimize their side effects. Heat-treated Korean ginseng (*Panax ginseng* C.A. Meyer) is one example of this. Sung et al reported that heat treatment of ginseng induces structural changes in ginsenosides and enhances the stability of ginseng saponins [14]. When the structural transformation of ginsenosides is induced through heat treatment, the minor ginsenosides with significant pharmacological and therapeutic activities, such as Rg2, Rg3, Rg5, Rh1, Rh2, Rh3, and Rh4, increases [15].

The heat treatment of ginseng causes chemical changes in its components and physical structure. In addition, various physical and chemical properties change during heat treatment depending on the temperature, time, number, and type of treatment [16]. However, there have been few reports on the changes in functional components and bioactivities of crude ginseng saponin (GS) induced by heat treatment.

The aim of this study is to confirm the effects of heat treatment on the ginsenoside composition and neuroprotective activities of GS, and to assess the clinical applicability of heat-treated crude ginseng saponin (HGS) for treating and preventing degenerative brain diseases.

## 2. Results

### 2.1. Changes in the Ginsenoside Compositions of Crude Ginseng Saponins Induced by Heat Treatment

The GS used in this study was made from ginseng 70% ethanol extract with GSH-20 resin, and subjected to heat treatment at 90 °C for 1 or 2 h. The heat-treated samples are referred to as HGS and the sample before heat treatment is referred to as NGS. Changes in the ginsenoside compositions of GS during heat-treatment are shown in Figure 1 and Table 1. As the heating time increased, the peaks of the major ginsenosides decreased, while those of the minor ginsenosides increased. In NGS, the ginsenosides Rg1, Re, Rb1, and Rb2 were the main components. However, the contents of these major ginsenosides decreased with heat treatment. On the other hand, saponins, such as the ginsenosides F4, Rh4, Rg3(S), Rg3(R), Rk1, and Rg5, were not initially detected in NGS, although their contents increased in proportion to heat treatment time. Ginsenoside Rh2 was produced only by the 2 h heat treatment, and F4 showed the highest content.

### 2.2. Protective Effects of HGS on Glutamate-Induced PC12 Cell Injury

In a previous study, to confirm the toxicity by GS, PC12 cells were treated with GS and HGS extracts at the concentrations of 5, 10, and 20 μg/mL, and then an MTS assay was performed. It was confirmed that the HGS extract had no cell toxicity even at 20 μg/mL. Based on these results, the concentration of 20 μg/mL, which does not affect cell viability, was used as the final concentration. PC12 cells were treated with GSs, including NGS, HGS1, and HGS2, and cell viability was measured by MTS assay (Figure 2A). None of the GSs affected cell viability at 20 µg/mL. To determine the effect of GSs on PC12 neuronal apoptosis induced by glutamate, cultured PC12 cells were pretreated with GSs (20 µg/mL) for 1 h, followed by treatment with 15 mM glutamate for 48 h. Cell viability decreased to 50% of the non-treated control group following glutamate treatment, and NGS, HGS1, and HGS2 treatment significantly increased viability to 80.26 ± 2.89, 82.95 ± 4.91, and 93.63 ± 4.14% of the non-treated control group, respectively (Figure 2B). HGS inhibited glutamate-induced PC12 cell apoptosis, and heat treatment enhanced its neuroprotective activity.

### 2.3. Effect of CGs on Glutamate-Induced ROS Production in PC12 Cells

DCFDA fluorescence staining intensity was measured to investigate the effect of GSs on glutamate-induced ROS formation in PC12 cells. ROS levels were significantly increased in glutamate-treated cells, but were reduced by co-treatment with GSs (Figure 3). ROS production was 1.7-fold higher in the glutamate-treated than in the untreated control group, and was inhibited 1.3-, 1.2-, and 1.1-fold by NGS, HGS1, and HGS2, respectively. HGS1 reduced glutamate-induced ROS formation more effectively than NGS, but HGS2 was the most effective. Overall, GS inhibited glutamate-induced ROS formation in PC12 cells, and heat treatment enhanced its activity.

### 2.4. Effect of HGSs on Antioxidant Protein Expression in Glutamate-Treated PC12 Cells

Western blotting was performed to investigate the effect of HGSs on the expression of the antioxidant protein nuclear factor E2-related factor 2 (Nrf2) and the intracellular antioxidant enzymes HO-1, SOD, CAT, and GPx in glutamate-treated PC12 cells. Nrf2 is a transcription factor that regulates antioxidant gene expression. As shown in Figure 4, the expression of Nrf2 was increased by HGSs compared to the glutamate-treated group (Figure 4A). Additionally, the expression levels of HO-1, SOD, CAT, and GPx, which are all targets of Nrf2, were upregulated by NGS, and even more so by HGSs, indicating that heat treatment enhanced the antioxidant properties of GS (Figure 4B).

### 2.5. Effect of HGSs on Apoptotic Protein Expression in Glutamate-Treated PC12 Cells

We found that the Bax/Bcl-2 ratio was increased by glutamate in PC12 cells, while the Bax/Bcl-xL ratio was unchanged. However, both ratios were reduced by GSs, and HGSs were more effective than NGS, indicating GS has anti-apoptotic properties; heat treatment enhanced these properties (Figure 5A). The apoptosis-related proteins cytochrome C, caspase-9, and caspase-3 showed similar patterns. The expression levels of these proteins were upregulated by glutamate; this effect was inhibited by GSs and HGS was more effective than NGS, confirming the protective effect of HGSs against glutamate-induced apoptosis (Figure 5B). These results suggest that HGSs exert a neuroprotective effect by inhibiting mitochondrial apoptosis.

### 2.6. Effect of HGSs on MAPK Signaling in Glutamate-Treated PC12 Cells

We investigated the effect of HGSs on MAPK signaling in glutamate-treated PC12 cells. HGSs inhibited the phosphorylation of JNK and p38, but not ERK, in glutamate-treated PC12 cells. The phosphorylation of JNK was especially increased by glutamate; this effect was suppressed by GSs, with HGS2 exerting the most dramatic effect (Figure 6). These data demonstrate that HGSs can protect PC12 neurons from glutamate-induced neurotoxicity by inhibiting JNK or p38 activation during MAPK signaling.

## 3. Discussion

The ginsenosides Rd, Re, and Rg3 may be useful to treat Alzheimer’s disease and improve the cognitive function of Alzheimer’s patients, as these compounds can pass through the blood–brain barrier [17]. This study confirmed the protective effect of HGS against glutamate-induced neurotoxicity, highlighting its possible clinical application for the prevention of degenerative brain diseases. Glutamate is an essential neurotransmitter in the central nervous system. However, when glutamate secretion is excessive, excitotoxicity is induced and nerve cells are killed due to oxidative damage [18]. The PC12 cells used in this study are derived from adrenal medulla chromaffin cells and express many enzymes involved in glutamate biosynthesis, metabolism, and uptake. PC12 cells are a model system for sympathetic ganglion-like neurons, widely used to study glutamate-induced apoptosis and assess the effects of neuroprotective substances [19]. A protective effect of the ginsenoside Rg2 against glutamate-induced damage in PC12 cells has previously been reported [20].

The saponin component in ginseng is converted into a new compound not produced by heat or acid treatment, or enzymatic hydrolysis. In ginseng, major ginsenosides are converted into minor ginsenosides by heat treatment. Traditionally, red ginseng is manufactured by steaming at 98–100 °C for 2–3 h, and there are reports that it has greater pharmacological effects than ginseng that has not been steamed [21]. Kim et al. reported that the contents of major ginsenosides, such as Rg1, Re, Rb1, Rc, Rb2, and Rd, decreased when steaming was repeated, and the content of Rg3, a minor ginsenoside, increased [22,23]. Nam et al. reported that the contents of the ginsenosides Rb1, Rb2, Rc, Rd, Re, Rf, and Rg1 decreased, and those of Rg3(S), Rg3(R), and Rk1 increased, in steamed ginseng [24]. In the present study, the contents of the ginsenosides F4, Rk3, Rg4, Rg3(S), Rg3(R), Rk1, and Rg3 converted from the major ginsenosides by heat treatment were increased in GS with longer heat treatment (Table 1). The ginsenosides Rb1, Re, and Rg1 gradually decreased according to the heat treatment time, whereas the contents of Rg2(S) and Rc increased significantly after 2 h of heat treatment. These results are similar to the change in ginseng seen in previous studies by steaming, suggesting that thermal processes can induce changes in ginsenoside composition and improve neuroprotective activities in GS.

The limitation of this study is that it is difficult to accurately describe which specific ginsenoside is effective because the correlation between the effects of each ginsenoside component of GS was not analyzed. However, in previous studies, the neuroprotective effect of ginsenoside was revealed in several experimental studies, suggesting that the HGS component also protects against glutamate-induced neurotoxicity in neurons.

Nrf2/ARE signaling in the brain may be an important neuroprotective target [25,26]. Calkins et al. demonstrated that Nrf2 plays an important protective role against neurotoxicity, and that Nrf2-mediated ARE transcription is a potential strategy for the prophylactic treatment of neurodegenerative diseases, such as Huntington’s disease [27]. In a study by Ye et al., the ginsenosides Rg1 and Rb1 exhibited neuroprotective effects by inducing HO-1 antioxidant enzymes via the activation of Nrf2/ARE signaling [28]. In the present study, the upregulation of the target proteins of Nrf2 by GS was enhanced in proportion to the heat treatment time (Figure 4). The expression of the antioxidant proteins HO-1, SOD2, CAT, and GPx1 in vivo was increased by NGS, and further increased by HGSs, compared to glutamate alone. These data show that Nrf2 activation by HGSs promotes the expression of antioxidant enzymes, such as HO-1, SOD2, GPx, and catalase, and inhibits intracellular ROS, resulting in neuroprotective effects.

Pro-apoptotic signals initially activate distinct signaling pathways but eventually converge on a common mechanism driven by the caspase family [29]. Bcl-2 and Bcl-XL inhibit apoptosis, whereas Bax induces the release of cytochrome c into the cytoplasm, resulting in apoptosis. The activation of procaspase 3 occurs after the cleavage of caspase-8, and cleaved caspase-3 directly promotes apoptosis [30,31]. In the present study, GS treatment inhibited apoptosis by increasing the Bax/Bcl-2 and Bax/Bcl-xL ratios, downregulating cytochrome c, caspase-9, and caspase-3. Overall, heat treatment improved the anti-apoptotic properties of GS (Figure 5).

Inflammatory cytokines and heat shock can trigger JNK and p38 signaling, and ERK signaling is sensitive to oxidative-stress-mediated neuronal cell death [32,33]. Glutamate-induced ROS in cultured neuronal cells activate various cellular signaling pathways, including pro-apoptotic stress-responsive JNK and p38 MAP kinase [34,35]. Ginseng compounds were found to induce anti-apoptotic activity through the modulation of caspase-3 expression and Erk and Akt phosphorylation [36,37]. Exposure to glutamate stimulation activates c-Jun N-terminal kinase (JNK), extracellular-signal-regulated kinase (ERK), and p38 MAPK by phosphorylation at locations that control the activation of the NF-κB signaling pathway. As a result, increased glutamate stimulates aging of neurons and microglia, leading to a vicious cycle that promotes neuroinflammation and neurodegeneration. Excess glutamate may act as a contributor to neurodegeneration, and reduced glutamate-induced Ca21 influx by ginsenosides may explain the neuroprotective effect observed in the present in vitro study. In the present study, we confirmed the effect of HGSs on MAPK signaling in glutamate-treated PC12 cells. HGSs inhibited the phosphorylation of JNK and p38, but not ERK, indicating that HGSs specifically suppress the activation of the JNK/p38 pathway (Figure 6). In other words, the anti-apoptotic properties of HGSs are exerted via the inactivation of the JNK/p38 pathway. Overall (Figure 7), our results show that HGS has excellent neuroprotective activity, suggesting that it has potential as a constituent of ginseng-derived neuroprotective pharmaceuticals.

## 4. Materials and Methods

### 4.1. Preparation of GS

Roots of ginseng (*Panax ginseng* C.A. Meyer.) harvested in Geumsan, Chungcheongnam-do in 2018 were air-dried at 55 °C for 72 h and pulverized into powder. The powder (10 kg) was extracted twice for 10 days in 60 L of 70% ethanol, and after filtration, the extract was vacuum-evaporated and freeze-dried as a ginseng extract (yield, 30%). GS was separated from the extract using a separation and purification system (Isolera Accelerated Chromatographic Isolation System; Biotage, Stockholm, Sweden). After filling the 500 mL column with GSH-20 resin, 100 mL of ginseng extract solution diluted with 3 L of distilled water and 25% ethanol was injected into the column. Next, 3 L of 95% ethanol was injected to create a saponin layer, which was concentrated using a vacuum condenser and lyophilized as GS (yield, 2.7%).

### 4.2. Preparation of HGS

The GS was heat-treated as follows. The heat-treatment was performed within a maximum of 2 h in consideration of quality changes, such as browning and off-flavor caused by excessive heating. A 200 mg/mL solution of saponin was sealed with a microcapillary glass pipette (Kimble Chase Life Science and Research Products LLC, Vineland, NJ, USA) and placed in a temperature-controlled water bath (Changshin Science Co., Seoul, Korea) at 90 °C for 0, 1, or 2 h (non-treated crude saponin (NGS), HGS1, and HGS2, respectively). The treated samples were diluted with 50 mg/mL 75% EtOH for liquid chromatography analysis, and with 100 mg/mL 50% DMSO for bioactivity analysis. Samples for liquid chromatography analysis were diluted to 5 mg/mL and filtered with a 4.5 μm membrane filter before use.

### 4.3. High-Performance Liquid Chromatography

Ginsenoside standards (Rg1, Re, Rf, Rb1, Rc, Rg2(S), Rh1, Rg2(R), Rb2, Rb3, F1, Rd, Rg6, F2, F4, Rk3, Rh4, Rg3(S), Rg3(R), Rk1, Rg5, and Rh2) were obtained from Ambo Institute (Seoul, Republic of Korea) and Chengdu Biopurify Phytochemicals Ltd. (Chengdu, Sichuan, China). HPLC-grade water and acetonitrile (ACN) were purchased from J.T. Baker (Phillipsburg, NJ, USA). All other chemicals were of reagent grade. Methanol (5 mL) was added to the dried material, and the mixture was vortexed to obtain a standard solution. The solution was filtered through a membrane filter (0.45 μm). Ginsenosides were analyzed by high-performance liquid chromatography (1200 Series; Agilent Technologies, Santa Clara, CA, USA). Under separation conditions, the mobile phase comprised distilled water, solvent A, ACN, and solvent B, with a concentration gradient of 80% (based on solvent A; 5 min), 60% (35 min), 60% (45 min), and 80% (50 min). A Mightysil RP-18 GP column (250 × 4.6 mm, 5 µm i.d.; Kanto Chemical Co., Tokyo, Japan) was used, with a temperature of 40 °C and flow rate of 1 mL/min. The diode-array detection detector wavelength was 203 nm.

### 4.4. Cell Culture

Pheochromocytoma 12 (PC12) cells derived from rat adrenal glands were purchased from ATCC (Manassas, VA, USA) and used for the experiments. PC12 cells were cultured in Dulbecco’s modified Eagle medium (DMEM), supplemented with 10% fetal bovine serum (FBS) and 1% penicillin–streptomycin at 37 °C in a 5% CO_2_ incubator. The medium was replaced every 2 days for subculture. Cells from passages 5–10 were used in all experiments.

### 4.5. Cell Viability

Cell viability was evaluated by the 3-(4,5-dimethylthiazol-2-yl)-5-(3-carboxymethoxyphenyl)-2-(4-sulfophenyl)-2H-tetrazolium (MTS; Promega, Madison, WI, USA) assay. PC12 cells were seeded in 96-well plates (1 × 10^4^ cells/well) and treated with 20 µg/mL of the control, NGS, HGS1, or HGS2 for 48 h. To investigate the effect of GSs on glutamate-induced cytotoxicity, PC12 cells were pretreated for 1 h with 20 µg/mL of the control, NGS, HGS1, or HGS2, and cytotoxicity was induced by treatment with glutamate (15 mM) for 48 h. After treatment with MTS and incubation for 1 h, absorbance at 490 nm was measured using a multi-plate reader (BioTek Instruments, Inc., Winooski, VT, USA).

### 4.6. Measurement of Intracellular Reactive Oxygen Species

Intracellular reactive oxygen species (ROS) generation was measured by a modified dichloro-dihydro-fluorescein diacetate (DCFH-DA) method. PC12 cells were cultured with the control or GSs in black 96-well plates at a density of 1 × 10^4^ cells/well. After 24 h, the cultured cells were treated with 15 mM glutamate in SFM for 20 min followed by 20 μM DCF-DA in a serum-free medium for 40 min. The cells were washed and 100 μL Dulbecco’s phosphate-buffered saline was added to each well. Fluorescence was measured with a multi-plate reader at 485 nm/535 nm (excitation/emission).

### 4.7. Protein Extraction

PC12 cells were pretreated with the control or GSs (20 μg/mL), followed by glutamate (15 mM/mL), for 24 h. Whole PC12 cell lysates were prepared in a radioimmunoprecipitation (RIPA) buffer (GenDEPOT, Katy, TX, USA) containing a protease and phosphatase inhibitor cocktail (GenDEPOT). Cell lysates containing equal amounts of protein were prepared using the Bradford assay. Proteins were mixed with 5× loading buffer and boiled for 10 min.

### 4.8. Western Blot Analysis

Protein samples were separated by 10% sodium dodecyl sulfate-polyacrylamide gel electrophoresis and transferred onto a polyvinylidene fluoride membrane (Millipore, Darmstadt, Germany). The membrane was washed three times and blocked by soaking in Tris-buffered saline + Tween (TBS-T) buffer (50 mM Tris-HCl (pH 7.5), 150 mM NaCl, and 0.1% Tween 20) containing 5% bovine serum albumin (GenDEPOT) for 30 min. The membrane was incubated with primary antibodies (1:1000) overnight at 4 °C. Then, the membrane was washed three times and incubated with the appropriate secondary antibody (1:2000) for 1 h at room temperature. All primary and secondary antibodies (β-actin, Nrf2, HO-1, SOD2, catalase, Gpx-1, Bax, Bcl/Bcl-xL, JNK/p-JNK, ERK/p-ERK, and p38/p-p38) were obtained from Abcam (Cambridge, UK) and Cell Signaling Technology (Beverly, MA, USA). Protein expression was visualized using enhanced chemiluminescence reagent (Bio-Rad, Hercules, CA, USA) according to the manufacturer’s instructions. Quantitative analysis was conducted using ImageJ software (version 1.52a for Windows; NIH, Bethesda, MD, USA).

### 4.9. Statistical Analysis

All experiments were repeated three times, and results are presented as the mean ± standard deviation (SD) of independent measurements. To increase the validity and reliability of the data scale and to confirm the normality of the data, the SPSS/PC 21.0 program was used to check the average value, standard deviation, skewness, and kurtosis. Statistical analysis (a one-way ANOVA and Duncan’s multiple range test) was performed using the Prism software (ver. 5.02; GraphPad Software, San Diego, CA, USA). *p*-values < 0.05 were considered to indicate statistical significance.

## 5. Conclusions

This study confirmed the effects of thermal processes on ginsenoside composition and neuroprotective activities in GS, and the neuroprotective mechanism of HGS against glutamate toxicity in neuronal PC12 cells. Heat treatment enhanced the neuroprotective effects of GS by increasing minor ginsenosides. HGS also protected PC12 cells against glutamate-induced oxidative stress by upregulating Nrf2-mediated antioxidant signaling and downregulating MAPK-mediated apoptotic signaling. Overall, our results show that HGS has excellent neuroprotective activity, suggesting that it has potential as a constituent of ginseng-derived neuroprotective pharmaceuticals.

## Figures and Tables

**Figure 1 ijms-24-07223-f001:**
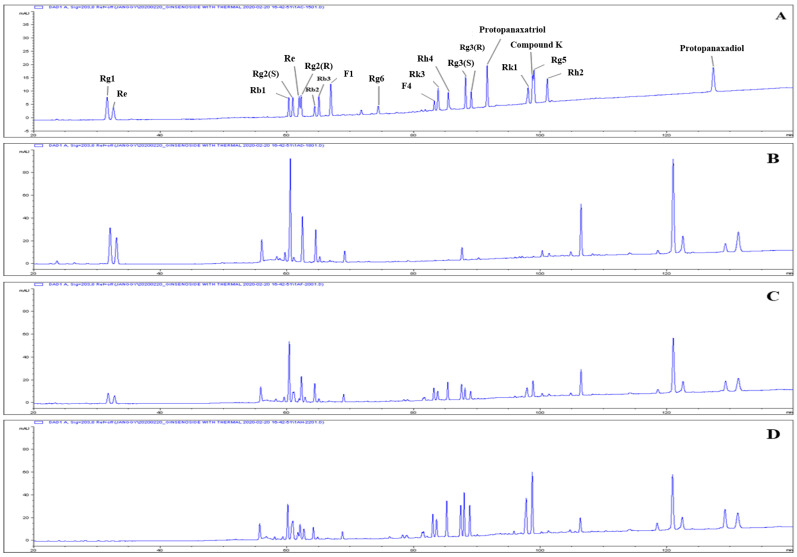
Typical chromatograms of ginseng crude saponin fraction with heat treatment. Samples: (**A**) standards; (**B**) control; (**C**) 90 °C—1 h; and (**D**) 90 °C—2 h.

**Figure 2 ijms-24-07223-f002:**
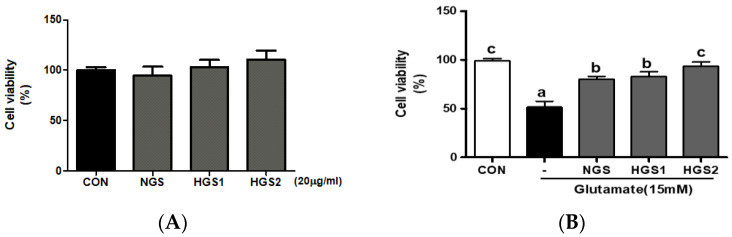
Inhibitory effect of HGS on glutamate-induced oxidative stress in PC12 cells. (**A**) PC12 cells were treated with HGS (20 μg/mL) and the control (0.2% DMSO) for 48 h. (**B**) PC12 cells were treated with HGS (20 μg/mL) and then with 15 mM glutamate for 48 h. Significance was determined by a one-way ANOVA with Tukey’s post hoc multiple comparison tests. Data are means ± standard errors of the mean (SEM). The values marked by different letters in a, b and c were found to differ significantly at *p* < 0.05 by Duncan’s multiple range test.

**Figure 3 ijms-24-07223-f003:**
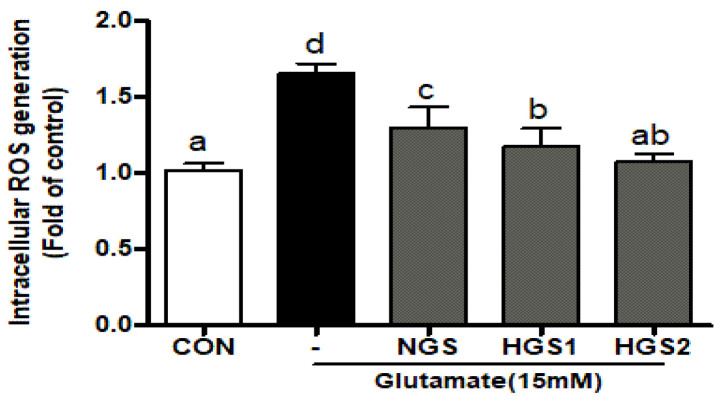
Inhibitory effect of HGS on glutamate-induced oxidative stress in PC12 cells. The cells were treated with HGS (20 μg/mL) or the control (0.2% DMSO) for 24 h and stimulated with glutamate (10 mM) for 4 h. ROS generation in PC12 cells was observed by fluorescence microscopy. Absorbance was measured after the DCF-DA staining of cells treated with glutamate or glutamate and HGS. Data are means ± standard errors of the mean (SEM). Significance was determined by a one-way ANOVA with Tukey’s post hoc multiple comparison test; The values marked by different letters in a, b, c and d were found to differ significantly at *p* < 0.05 by Duncan’s multiple range test.

**Figure 4 ijms-24-07223-f004:**
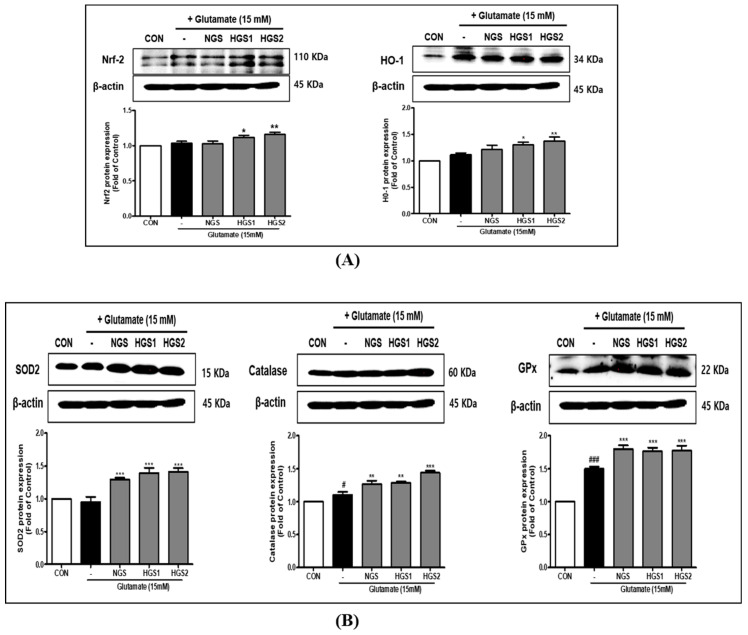
Effect of HGS on the expression of antioxidant enzymes in PC12 cells. Cells were treated with HGS (20 μg/mL) or the control (0.2% DMSO) for 1 h followed by glutamate (15 mM) for 24 h. Proteins were analyzed by Western blotting using β-actin as the loading control. Nrf2, HO-1, SOD2, CAT, and GPx levels in PC12 cells. Data are means ± standard errors of the mean (SEM). A one-way ANOVA determined significance with Tukey’s post hoc multiple comparison test. # *p* < 0.1 and ### *p* < 0.001, significance compared with the control (white bar). * *p* < 0.1, ** *p* < 0.01, and *** *p* < 0.001, significance compared with the glutamate-treated cells (black bar).

**Figure 5 ijms-24-07223-f005:**
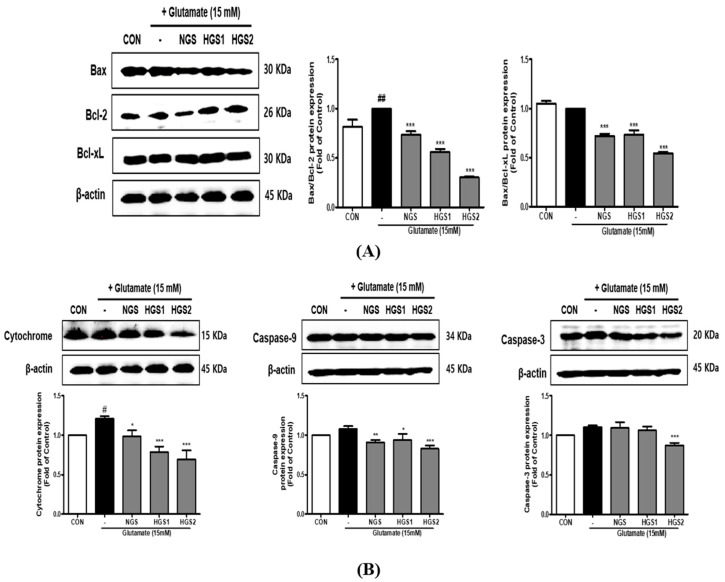
Effect of HGS on glutamate-induced Bcl-2 family protein expression in PC12 cells. PGS protects PC12 cells from glutamate-induced apoptosis. Cells were treated with HGS (20 μg/mL) or the control (0.2% DMSO) for 1 h followed by glutamate (15 mM) for 24 h. Proteins were analyzed by Western blotting using β-actin as the loading control. (**A**) Bax/Bcl-xL and Bax/Bcl-2 protein expression in the mitochondria. (**B**) Cytochrome c and caspase-9 and -3 expression in mitochondria. Data are means ± standard errors of the mean (SEM). Significance was determined by a one-way ANOVA with Tukey’s post hoc multiple comparison test. # *p* < 0.1, ## *p* < 0.01 significance compared with the control (white bar). * *p* < 0.1, ** *p* < 0.01, and *** *p* < 0.001, significance compared with the glutamate-treated cells (black bar).

**Figure 6 ijms-24-07223-f006:**
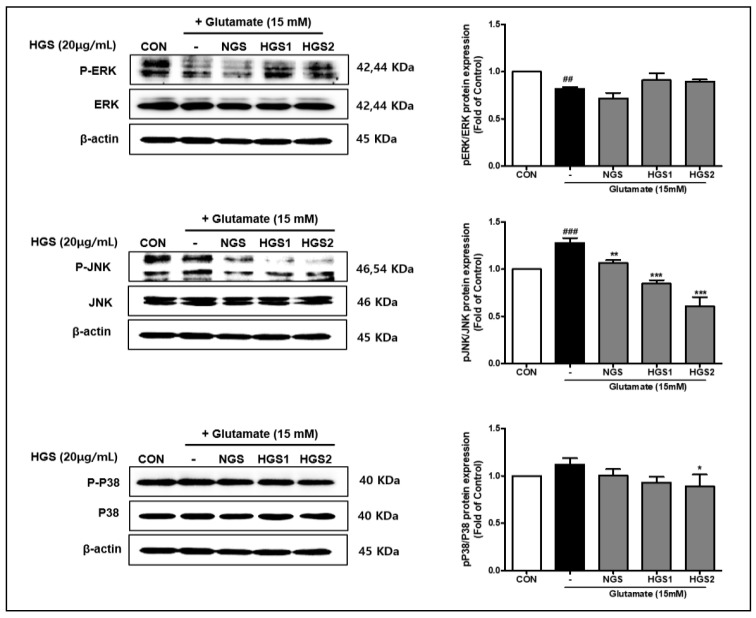
Effect of HGS on MAPK expression in glutamate-induced PC12 cells. Cells were treated with HGS (20 μg/mL) or the control (0.2% DMSO) for 1 h followed by glutamate (15 mM) for 24 h. Proteins were analyzed by Western blotting using β-actin as the loading control. Data are means ± standard errors of the mean (SEM). Significance was determined by a one-way ANOVA with Tukey’s post hoc multiple comparison test. ## *p* < 0.01 and ### *p* < 0.001, significance compared with the control (white bar). * *p* < 0.1, ** *p* < 0.01, and *** *p* < 0.001, significance compared with the glutamate-treated cells (black bar).

**Figure 7 ijms-24-07223-f007:**
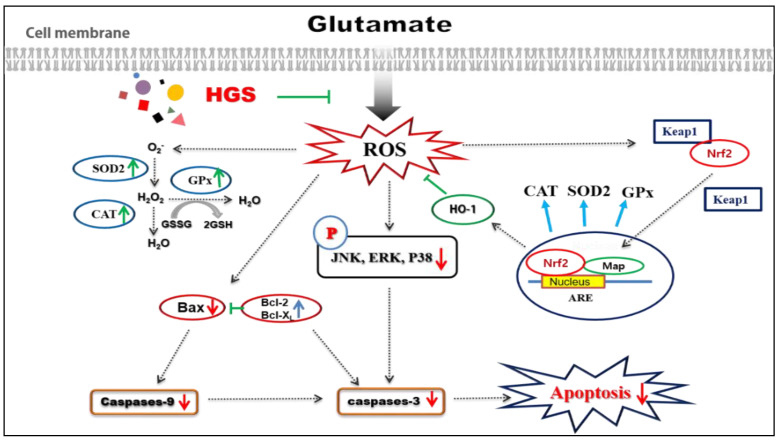
Schematic of the mechanisms of the amelioration by HCS of glutamate-induced neuronal cell death.

**Table 1 ijms-24-07223-t001:** Ginsenoside composition of crude ginseng saponin. Unit: mg/g.

Ginsenoside		Sample	
	NGS	HGS1	HGS2
Rg1	33.13 ± 0.59	22.05 ± 0.48	16.14 ± 0.99
Re	55.35 ± 2.61	38.01 ± 2.86	25.98 ± 2.10
Rb1	130.89 ± 3.36	124.43 ± 4.60	117.51 ± 0.86
Rg2(S)	6.92 ± 0.60	7.07 ± 0.42	16.00 ± 0.77
Rc	N.D.	1.78 ± 0.20	6.56 ± 0.36
Rg2(R)	56.24 ± 0.87	46.43 ± 1.66	46.98 ± 2.71
Rb2	66.48 ± 3.50	61.52 ± 4.36	63.82 ± 4.37
Rb3	6.23 ± 0.19	5.62 ± 0.18	4.58 ± 1.51
F1	N.D.	N.D.	N.D.
Rg6	N.D.	N.D.	N.D.
F4	N.D.	25.74 ± 0.42	44.07 ± 3.83
Rk3	N.D.	6.95 ± 0.41	13.83 ± 0.78
Rh4	N.D.	19.58 ± 1.55	34.63 ± 3.28
Rg3(S)	N.D.	6.04 ± 0.68	13.02 ± 0.38
Rg3(R)	N.D.	9.44 ± 0.41	17.59 ± 2.14
PPT	N.D.	N.D.	N.D.
Rk1	N.D.	6.72 ± 0.45	13.15 ± 0.64
Compound k	N.D.	N.D.	N.D.
Rg5	N.D.	7.84 ± 0.63	17.15 ± 1.32
Rh2	N.D.	N.D.	2.56 ± 0.26
PPD	N.D.	N.D.	N.D.
Total	355.24 ± 3.38	389.24 ± 4.34	453.58 ± 8.07

Data are expressed as mean ± standard deviation of triplicate samples. N.D., not detected.

## Data Availability

Not applicable.

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
