# Peer review of "Heat Treatment Enhances the Neuroprotective Effects of Crude Ginseng Saponin by Increasing Minor Ginsenosides"

_ijms, 2023, doi:10.3390/ijms24087223_

Round 1
Reviewer 1 Report
Abstract
Line 11 immune- sugar- ... need to be improvised
1. Introduction
Line 45 PPD & PPT , the spelling
Line 54 unclear and ambiguous statement what enchance the stability of ginseng saponins
2. Material and Methods
Part 2.3 number of technical/ biological replicates should be indicated in the writeup
Line 70 to prepare samples (incomplete and unclear statements about what is samples meant here)
Line 71 tenses for ground
Part 2.8 Western blot analysis: the paragraph is about Sprague Dawley treatment with AK diet. This is completely contradicting and not reflected in the title of the method (2.8). The method on this Sprague dawley treatment is not reflected in the result completely.
3. Results
Resolution for figure 1 (especially the labeling in Fig1.A need to be improvised
3.2 Wrong abbreviation used (CG)
4. Discussion
Discussion did highlight the changes after the heat treatment on saponin in MAPK signaling, apoptosis. However, the underlying mechanism why heat treatment would be able to enhance it is not strongly related in the writeup (such as the Ginesoid contents with signalling) . Besides, the heat treatment only done for 1, 2 hr , however, the rationale why 1 , 2 hrs being selected is not clearly stated. Besides, how would be the implication or application of the findings in this study is not well discussed.
There is no Figure 6 in writeup as mentioned in Fig. 6.
Reviewer 2 Report
The present study aimed to investigate the enhancing effects of heat treatment on the beneficial effects of Ginseng in neuronal cells, which is of some interest. However, there are some concerns from the reviewer:
1. For the cell viability treatment, only one concentration for three types of GS was employed to PC12 cells, which was not convincing for the results because the effects of lots phytochemicals appear to be dose-dependent. The authors need to justify why only one concentration has been selected and compare the actions between three type of GS at various concentrations.
2. Glutamate was used to induce the oxidative stress damage in PC12 cells to investigate the protective effects of different types of GS. These results have been over-interpreted as "neurodegenerative diseases". Although the oxidative stress do have crucial role in the pathogenesis of the neurodegenerative diseases, no diseases model were used to evaluate the beneficial effects of GS.
3. In the statistical analysis, please provide further information about the statistical package used, normality test. It seems that the authors used parametric tests, does this mean that the data were normally distributed? Which test was used to do so? Please provide the result of checking.
4. For the presentation of the figures, please provide the total number of the experiment.
5. Language editing is encouraged.
6. Actually, only cell model was employed in the present study, which is less convincing.
7. What are the limitations of this study?
Round 2
Reviewer 1 Report
The authors had done tremendous amendments to the comments. No further comments .